# ECONAGENTBENCH: ECONOMIC BENCHMARKS FOR LLM AGENTS IN UNKNOWN ENVIRONMENTS

## ABSTRACT

We develop benchmarks for LLM agents that act in, learn from, and strategize in unknown economic environments, the specifications of which the LLM agent must learn over time from deliberate exploration. Our benchmarks consist of decision-making tasks derived from key problems in economics. To forestall saturation, the benchmark tasks are synthetically generated with scalable difficulty levels. Overall, our benchmarks assess the abilities of LLM agents in tackling complex economic problems in procurement, scheduling, and pricing—applications that should grow in importance as such agents are further integrated into the economy.

## 1 INTRODUCTION

Organizations increasingly delegate parts of their economic decision-making to LLMs.[1] Over the last year, LLMs have sufficiently matured such that the potential for *LLM agents* is increasingly realizable, which further promotes such delegation.[2] Economic decisions—such as on procurement, scheduling, and pricing—are often made in uncertain environments and require trial and error. However, the performance of LLM agents in such environments is not a main focus of existing benchmarks.

To address this question, we introduce **EconAgentBench**: an array of benchmarks for LLM agents that act in, learn from, and strategize in unknown environments, the specifications of which the LLM agent must learn over time from deliberate exploration. Specifically, we develop benchmarks for three core economic tasks: procurement, scheduling, and pricing. We employ each of the benchmarks at three different difficulty levels: BASIC, MEDIUM, and HARD. The benchmarks consist of synthetic environments and can therefore be quickly scaled in size and complexity even beyond these three levels as LLM capabilities continue to progress.

Our contributions are as follows:

1. We propose **EconAgentBench**, an array of benchmarks measuring the capabilities of LLM agents in three key economic settings: procurement, scheduling, and pricing.

2. We construct the benchmark environments in a way that allows for **difficulty scaling** to arbitrarily high difficulty levels (to forestall saturation), and experimentally validate this technique.

3. We evaluate the performance of a diverse array of LLM agents (including LLM agents based on cutting-edge models GPT-5 and Gemini 2.5 Pro) and demonstrate that analyzing the way LLM agents tackle our benchmarks can give rise to **economically meaningful insights** regarding mechanisms underlying observed differences in benchmark scores.

---

[1]Handa et al. (2025) analyze usage data of Claude.ai, and find that 5.9% of conversations relate to business or finance.

[2]In an April 2025 appearance on Bloomberg Technology, Visa CEO Ryan McInerney describes Visa's vision for "[LLM] agents to buy on your behalf" (Bloomberg, 2025). Constantz (2024) reports on similar such LLM agent integration and delegation at McKinsey, and also on the rise of commercial-grade LLM agent releases by companies such as OpenAI and Salesforce. Reed (2024) report the adoption of generative-AI-driven pricing by airlines including Delta and Virgin Atlantic.

## 2 RELATED WORK

**Economic benchmarks for LLMs.** Other benchmarks measuring the economic capabilities of LLMs include STEER (Raman et al., 2024), STEER-ME (Raman et al., 2025), and VendingBench (Backlund & Petersson, 2025). STEER and STEER-ME are Q&A benchmarks; by contrast, EconAgentBench evaluates LLM agents in various economic environments. VendingBench (concurrent work) consists of a specific environment simulating a single economic task—operating a vending machine—whereas EconAgentBench evaluates LLM agents on a diverse array of economic tasks (procurement, scheduling, and pricing) in environments that can be found throughout the economy.

**Benchmarks for frontier LLMs.** Two key problems in benchmark design and maintenance are *saturation* (see e.g. Phan et al., 2025) and *data contamination* (see e.g. OpenAI, 2024; Jose, 2024). Recent benchmarks such as FrontierMath, ARC-AGI, HLE, and NYT-Connections address saturation by relying on human experts to craft difficult questions, and data contamination by only partially releasing the benchmark Glazer et al. (2024); Chollet et al. (2025); Phan et al. (2025); Loredo Lopez et al. (2025). We share the goal of creating hard and future-proof benchmarks, and adopt the approach of using synthetic instance generation (see, e.g., Valmeekam et al., 2023). This allows for scaling the difficulty of benchmark tasks as well as making the benchmark code publicly available.

**LLMs for Multi-turn RL.** Ma et al. (2024) categorize multi-turn incomplete-information LLM agent benchmarks into four categories: *embodied* (physical instructions), *web* (browser usage), *tool* (measuring the ability to usefully call external functions), and *game* (video game-style environments).[3] Our benchmarks do not neatly fit into any of these four categories. Rather, our benchmarks, which simulate realistic usage of LLMs in economic scenarios, might fall into a fifth *optimization* category. Optimization problems are well-suited for multi-turn LLM agent benchmarks because they are naturally equipped with a fine-grained progress metric (see Ma et al., 2024, for general discussion of the importance of fine-grained progress metrics). Other multi-turn optimization environments that may be fruitful for future work include multi-armed bandit settings and assortment optimization (see, e.g., Krishnamurthy et al., 2024).

## 3 BENCHMARK DESIGN

We design environments that simulate three core economic tasks: procurement, scheduling, and pricing. In each setting, the LLM agent acts in the environment for 100 periods. Each period culminates with the LLM agent taking a single action (e.g., setting a price), after which the LLM agent receives feedback.[4] In all of our environments, there is a well-defined notion of an optimal action (in a given period), and a natural way to measure the relative quality of a non-optimal action (in that period).

In Section 3.1, we describe the "API" via which the LLM agent acts in the benchmark environment. In Section 3.2, we describe the architecture of the LLM agents we test. In Section 3.3, we provide the design details of the three benchmark environments. Finally, in Section 3.4, we conclude with a high-level overview of the key design features of EconAgentBench.

### 3.1 BENCHMARK INTERACTION METHOD

Rather than designing benchmark questions with which to query an LLM, we design benchmark *environments* in which an *LLM agent* must act (and is evaluated). LLM agent technology is nascent and there is currently no singular standard interaction protocol.[5] To ensure versatility and future-proofness of our benchmarks, we only require a lightweight interaction protocol using **tool use** (also referred to as function calling). We select this interaction method because it has rich precedent in the literature on agentic workflows (see, e.g. Schick et al., 2023) and is included in frontier LLMs as a built-in feature.

---

[3]See also Wang et al. (2023); Mialon et al. (2023); Xie et al. (2024); Ma et al. (2024); Liu et al. (2023); He et al. (2024); Zhou et al. (2023).

[4]In this sense, our environments can be viewed as POMDPs (see, e.g., Ma et al., 2024, for such framing).

[5]Example recent proposals include Anthropic's Model Context Protocol and Google's Agent2Agent protocol. See also Chan et al. (2025).

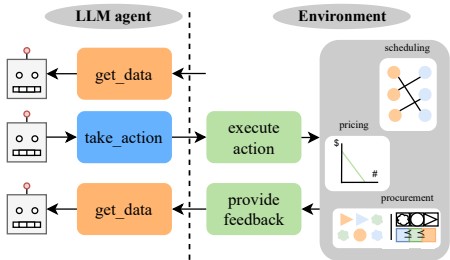

Figure 1: Illustration of how the LLM agent interacts with the benchmark environment. The LLM agent obtains information and takes actions via tool use (see Section 3.1). The environment performs computations based on the tools used and returns information (see Section 3.3).

Each economic environment is associated with a list of tools. There are two types of tools: *getter tools*, which return information about the environment, and *action tools*, which execute an action (e.g., setting a price). Table 1 lists the associated tools for each benchmark environment (further detail in Appendix F). When the LLM agent calls a getter tool, the relevant quantity is computed according to parameters of the underlying (synthetic) economic and returned; when the LLM agent calls an action tool, the underlying economic environment computes the consequences of that action and advances to the next period. See Figure 1 for an illustration.

Table 1: Overview of tools associated with each economic environment

| Environment | Getter tools | Action tool |
|---|---|---|
| Procurement | get_previous_purchase_data, get_equipment_information, get_budget, get_attempt_number | submit_purchase_plan |
| Scheduling | get_previous_attempts_data, get_worker_ids, get_task_ids, get_attempt_number | submit_assignment |
| Pricing | get_previous_pricing_data, get_product_ids, get_attempt_number | set_prices |

Accordingly, any LLM agent capable of using the tools listed in Table 1 can be evaluated using our benchmarks. While LLM agents interact with our environments using the above tools, we remark that LLM agents are not limited to using only these tools. For example, in this work we test LLM agents equipped with additional tools allowing for memory between periods (see Section 3.2). As the capacity for LLMs to use increasingly large sets of tools advances, one could imagine augmenting LLM agents with additional tools, e.g., a (secure) Python interpreter.

### 3.2 LLM Agent Architecture

For each frontier LLM that we test, we construct an LLM agent by equipping the LLM with tools that allow it to act in the benchmark environment as well as formulate and keep track of its plans. Each period is conducted in a single chat session.[6] At the start of each period, the LLM agent is given the same initial instructions and a list of tools it can use to interact in the economic environment. The tools include the environment-specific tools described in Table 1, as well as two additional *notes tools*, write_notes and read_notes, that allow the LLM agent to read and write notes to itself that persist between periods.[7] For further details on the functionality of the notes tools, see Appendix F. All LLMs are queried at temperature 1.

### 3.3 Economic Environments

We design three benchmark environments to simulate a broad array of key economic tasks. The environments we construct come in two forms: *stationary* and *non-stationary*.

---

[6]Our benchmarks thus require a relatively long context window, a condition satisfied by the LLMs we use.

[7]Equipping LLM agents (or workflows) with a sufficiently flexible memory module has been shown to be critical for their performance at economic (Fish et al., 2024) and agentic exploration (Krishnamurthy et al., 2024) tasks.

In the stationary environments (procurement and scheduling), the quality of an action does not depend on the period in which it is taken, and accordingly the LLM agent is scored based on the quality of its best or final action. In particular, to earn a perfect score in a non-stationary environment, it suffices for the LLM agent to identify and take an optimal action once.

In the non-stationary environments (pricing), the quality of an action changes over time according to a predictable pattern that the LLM agent must learn, and accordingly the LLM agent is scored based on its ability to consistently take high-quality actions after an initial exploration period. In particular, to earn a perfect score in a non-stationary environment, the LLM agent must take optimal actions many periods in a row, changing them appropriately as the environment changes.

### 3.3.1 PROCUREMENT

**High-level overview.** The LLM agent is given a list of prices for bundles of products (e.g., "$2 for 2 units of product A and 3 units of product B"), and a budget. Every period, the LLM agent proposes a purchase plan, and receives as feedback the quality of that purchase plan (determined by a simple, but unknown to the LLM agent, mathematical formula). The LLM agent's goal is to identify the best purchase plan within the budget.

**Environment.** There are $n$ products $A := \{a_1, \ldots, a_n\}$ with effectiveness scores $e_1, \ldots, e_n \in \mathbb{N}$. The products are partitioned into $k$ categories $A := A_1 \sqcup \cdots \sqcup A_k$ (where $\sqcup$ denotes disjoint union).

Given quantities purchased of each product $(z_1, \ldots, z_n) \in \mathbb{Z}_{\geq 0}^n$, the quantity of workers supported by these products is given by

$$f(z_1, \ldots, z_n) := \prod_{i=1}^k \left( \sum_{a_j \in A_i} e_j z_j \right)^{1/k}.$$

Thus, products within the same category are substitutes, and products across different categories are complements.

Products can be purchased through *deals*. There are three types of deals: *simple* (a bundle of products is assigned a per-copy price), *bulk only* (like simple, but requires purchasing at least some minimum number of copies), and *two-part tariff* (like simple, but in addition to the per-copy price there is also an upfront cost for the deal that is independent of the number of copies purchased). For further details see Appendix C.

**Task.** The LLM agent is given a budget $B > 0$ and a menu consisting of $m$ deals. It is asked to find the purchase plan of deals that maximizes the quantity of workers supported within the budget.

**Tools.** The LLM agent has access to the following tools: get_previous_purchase_data, get_equipment_information, get_budget, get_attempt_number, submit_purchase_plan. For further details see Appendix F.1.

**Feedback.** In each period, the LLM agent may propose a purchase plan. If the purchase plan exceeds the budget, the agent is informed that the plan is not feasible. Otherwise, the agent receives feedback on the quantity of workers supported by that purchase plan.

**Key Unknowns.** The LLM agent is not given the effectiveness scores $e_1, \ldots, e_n \in \mathbb{R}$, and must learn information about these weights indirectly from the feedback.

**Instantiation.** We set $n = 12$ and $k = 3$ for BASIC, $n = 30$ and $k = 5$ for MEDIUM, and $n = 100$ and $k = 10$ for HARD. The effectiveness scores $e_1, \ldots, e_n$ are sampled uniformly from $\{1, 2, 3\}$ for BASIC, $\{1, 2, \ldots, 5\}$ for MEDIUM, and $\{1, 2, \ldots, 20\}$ for HARD. For each difficulty level we set the menu size $m := n$ and we use equal category sizes $|A_1| = \cdots = |A_k| = n/k$. For details of menu generation see Appendix C.

**Success Metric.** Each experimental run is scored based on the quantity of workers supported by the best purchase plan the LLM agent proposed, normalized by the quantity of workers supported by the optimal purchase plan within budget $B$:

$$\frac{f(\text{LLM's quantities purchased of each product})}{\text{OPT}}.$$

### 3.3.2 SCHEDULING

**High-level overview.** The LLM agent is given a list of workers and tasks. The workers have preferences over the tasks, and the tasks have "preferences" over the workers (e.g., determined by how suitable a worker is for that task), but the LLM agent is not explicitly told any of these preferences. Every period, the LLM agent proposes an assignment of workers to tasks, and receives as feedback one or more "problems" with that assignment. The LLM agent's goal is to identify an assignment with no, or as few as possible, "problems."

**Environment.** There are $n$ workers $W := \{w_1, \ldots, w_n\}$ and $n$ tasks $T := \{t_1, \ldots, t_n\}$. Each worker $w_i$ has a complete strict preference order $\succ_{w_i}$ over tasks, and each task $t_i$ has a complete strict preference order $\succ_{t_i}$ over workers.

**Task.** The LLM agent is asked to find a *(perfect) matching* that is *stable*. A *matching* is a bijection $\mu : W \to T$. A worker-task pair $(w, t) \in W \times T$ is a *blocking pair* for a matching $\mu$ if $t \succ_w \mu(w)$ and $w \succ_t \mu(t)$, that is, $w$ and $t$ each prefer the other over their match in the matching. A matching is *stable* if it has no blocking pairs. The existence of a stable matching is guaranteed by Gale & Shapley (1962).

**Tools.** The LLM agent has access to the following tools: get_previous_attempts_data, get_worker_ids, get_task_ids, get_attempt_number, submit_assignment. For details about the precise functionality of these tools see Appendix F.2.

**Feedback.** In each period, the LLM agent may propose a matching. If the matching is stable, the experiment ends. Otherwise, the agent receives feedback in the form of $k$ randomly chosen blocking pairs (or all blocking pairs, if there are fewer than $k$).[8]

**Key Unknowns.** The LLM agent is not given the preferences of the tasks and workers $\succ_{w_i}$ and $\succ_{t_i}$, and must learn information about these preferences indirectly from the blocking-pair feedback.

**Instantiation.** We set $n = 10$ and $k = 1$ for BASIC, $n = 20$ and $k = 2$ for MEDIUM, and $n = 50$ and $k = 5$ for HARD. For each difficulty level, we randomly generate the preferences of the workers and tasks using the public scores model (Ashlagi et al., 2023). For details of preference generation see Appendix D.

**Success Metric.** Each experimental run is scored based on the quality of the final matching the LLM agent proposes,[9] according to the following formula:

$$1 - \frac{\text{\# blocking pairs in agent's final matching}}{\mathbb{E}_{\text{unif. random matching } \mu}[\text{\# blocking pairs in } \mu]}.$$

Note that the formula allows for negative scores if the LLM agent proposes a final matching that is worse than the uniform random baseline.

### 3.3.3 PRICING

**High-level overview.** The LLM agent is given a list of products. Every period, the LLM agent sets prices for those products, and receives as feedback the quantity sold and profit earned from each product (determined by a simple, but unknown to the LLM agent, mathematical formula). The LLM agent's goal is to set prices in a way that maximizes profits. Moreover, the market conditions change according to a predictable pattern, and to price optimally, the LLM agent must anticipate this pattern and price accordingly (e.g., learn to steadily increase or decrease prices).

---

[8] A stable matching can be computed in polynomial time based on this input, even if only one, adversarially chosen, blocking pair is returned (Bei et al., 2013; Emamjomeh-Zadeh et al., 2020).

[9] In the final period, the following additional instruction is included in the LLM prompt: *"**This is your final attempt.** This time, you should submit the highest quality assignment possible, that has the fewest problems."* This ensures that the LLM agent is evaluated based on a matching for which it was instructed to minimize the number of blocking pairs (mitigating the risk that it uses the final period to explore).

**Environment.** There are $n$ products $G := \{g_1, \ldots, g_n\}$ partitioned into $k$ categories $G := G_1 \sqcup \cdots \sqcup G_k$ (where $\sqcup$ denotes disjoint union). Given prices $p_1, \ldots, p_n$, the quantity demanded $q_i$ for the $i$th product $g_i$ in the $j$th category $G_j$ is given by a nested logit demand model (Berry, 1994):

$$q_i := M \frac{\exp(\frac{a_i - p_i/\alpha_i}{1-\sigma})}{D_j} \cdot \frac{D_j^{1-\sigma}}{\exp(\frac{a_0}{1-\sigma}) + \sum_{j' \in [k]} D_{j'}^{(1-\sigma)}},$$

where $D_{j'} := \sum_{g_k \in G_j} \exp(\frac{a_k - p_k/\alpha_k}{1-\sigma})$ for $j' \in [k]$. Here, $a_i$ is the quality of product $g_i$ (higher is better), $a_0$ is the quality of an outside option (higher means outside option more attractive), $\alpha_i$ determines the price sensitivity, $D_j$ is the market share of category $G_j$, $\sigma$ is the elasticity of substitution, and $M$ scales overall market share.

Given costs $c_1, \ldots, c_n$ of the products, the profit from good $g_i$ is $\pi_i := (p_i/\alpha_i - c_i)q_i$. The total profit is $\pi := \sum_{i=1}^n \pi_i$.

To make this pricing environment non-stationary, we vary the $\{\alpha_i\}_{i=1}^n$ parameters between periods, according to a predictable pattern that the LLM must learn. We consider two kinds of patterns: *linear shifts*, in which each $\alpha_i$ is increased or decreased by a constant step size in each period (the step sizes differ between products $i \in [n]$), and *periodic shifts*, in which each $\alpha_i$ varies according to a sinusoidal pattern (the frequency and phase are the same for all products $i \in [n]$, but the amplitudes may differ).

**Task.** The LLM agent is asked to set prices for the $n$ products in a way that maximizes total profit $\pi$.

**Tools.** The LLM agent has access to the following tools: get_previous_pricing_data, get_product_ids, get_attempt_number, set_prices. For details about the precise functionality of these tools, see Appendix F.3.

**Feedback.** At the end of each period, the LLM agent sets prices for the $n$ products. In the following period, the LLM agent is given as feedback the quantity sold and profit earned for each product, as well as total profit.

**Key Unknowns.** The LLM agent is not given the parameters $\{a_i\}_{i=1}^n, \{\alpha_i\}_{i=1}^n, a_0, \sigma, M$ that characterize the demand response (nor how they evolve, where applicable), and must learn information about these parameters indirectly from the feedback.

**Instantiation.** To scale the difficulty, we scale the number of products. We set $n = 1$ for BASIC, $n = 4$ for MEDIUM, and $n = 10$ for HARD. Across all difficulty levels, we set $\sigma = 0.5$ and $M = 100$. We sample the costs $c_i \sim \mathsf{Unif}([1, 10])$ and qualities $a_i \sim \mathsf{Unif}([2, 3])$ independently. For each product $i \in [n]$, its category membership is determined by sampling from a (right-)truncated geometric distribution $\mathsf{Geom}(0.2)$. To make the pricing environment non-stationary, we vary the $\{\alpha_i\}_{i=1}^n$ parameters with time according to a predictable pattern (either linear shifts or periodic shifts). For further details see Appendix E.

**Success Metric.** Each experimental run is scored based on the total profit earned in the last 50 periods, normalized by the total profit that would have been earned from pricing optimally in those periods:

$$\frac{\text{total profit } \pi \text{ from last 50 periods}}{\mathsf{OPT}}.$$

### 3.4 KEY DESIGN FEATURES

Each environment is **synthetically generated** according to an underlying economic model. Accordingly, it is possible to generate and test on arbitrarily many benchmark instances. Additionally, each environment is designed to allow for **scalable difficulty**—e.g. in scheduling, the difficulty can be increased by increasing the number of workers and tasks. In this work, we instantiate each economic environment at three different difficulty levels—BASIC, MEDIUM, and HARD—but it is possible

to generate instances at arbitrary difficulty levels. Finally, the difficulty of each benchmark task lies (partly) in that the LLM agent must operate in an **unknown environment**—e.g. in scheduling, the preferences of the workers and tasks are not given to the LLM agent, and can only be learned via deliberate exploration. Thus, it is not possible for any agent or algorithm, no matter how sophisticated, to consistently produce a perfect solution to a benchmark task in the first period. In this sense, a key feature of the benchmark environments we construct is not only that they simulate core economic tasks, but also that they test the ability for LLM agents to **reason under uncertainty** more generally.

## 4 BENCHMARK RESULTS

In this section, we demonstrate key features of our benchmarks by analyzing the performance of LLM agents based on an array of frontier LLMs at our EconAgentBench benchmarks. First, in Section 4.1, we validate our difficulty scaling approach by comparing the scores of a broad array of LLM agents across the difficulty levels BASIC, MEDIUM, and HARD. In Section 4.2, by measuring the performance of two additional, cutting-edge LLM agents, we show that despite improvements in LLM technology, our benchmarks are not saturated at the HARD difficulty level. Finally, in Section 4.3, we demonstrate that economic insights can be uncovered by analyzing the behavior of LLM agents beyond just their overall scores. For additional experimental details, including information on data collection timeframes and costs, see Appendix A.

### 4.1 VALIDATING DIFFICULTY SCALING: A BROAD COMPARISON

Our first goal is to assess the extent to which our proposed difficulty scaling technique is effective. To do so, we measure the performance of LLM agents based on Claude-3.5 Sonnet (20241022 version), Gemini 1.5 Pro (002 stable release), GPT-4o (20241120 version), GPT-4.1 (20250414 version), and o4-mini (20250416 version) on the three EconAgentBench environments (procurement, scheduling, and pricing). We instantiate each economic environment at three different difficulty levels—BASIC, MEDIUM, and HARD—and for each difficulty level, we randomly generate 12 instances and run all LLM agents for 100 periods on the same instances. The final benchmark score of each LLM agent is computed by averaging the scores of the individual runs.

The benchmark results are summarized in the upper part of Table 2. We observe that our approach for scaling the difficulty of benchmarks—increasing the instance size—is effective. For example, for all LLM agents and all three economic environments, scores on HARD instances are lower than scores on BASIC instances ($p < 0.05$, one-sided Welch's $t$-test).

### 4.2 NONSATURATION: EVALUATION ON CUTTING-EDGE MODELS

We additionally measure the performance of LLM agents based on GPT-5 (20250807 version) and Gemini 2.5 Pro (June 2025 version) on EconAgentBench at the HARD difficulty, for 100 periods, on the same instances as in Section 4.1. The lower part of Table 2 displays the results. We observe that GPT-5 emerges as the clear leader in the two stationary benchmark environments (procurement and scheduling), whereas perhaps surprisingly, GPT-4.1 achieves the highest score in pricing (the only non-stationary environment), closely followed by Gemini 2.5 Pro. This result highlights that our individual benchmarks measure different dimensions of skill in economic environments, and underscores the need for domain-specific benchmarks.

### 4.3 ECONOMIC INSIGHTS: INTER-MODEL COMPARISONS, IMPROVEMENT MECHANISMS

In this section, we examine the behavior—beyond merely the overall benchmark scores—of the LLM agents on EconAgentBench at the HARD difficulty level, with the goal of gaining economically meaningful insights regarding mechanisms underlying differences in scores. Specifically, in each of the three benchmark environments, we identify an action-quality metric and analyze it across LLM agents. For additional analysis of exploration in the procurement and scheduling environments, see Appendix B.

**Procurement.** To better understand differences in procurement scores, we study *budget utilization*—the proportion of purchase plans with cost between 95% and 100% of the budget, that is, a

Table 2: Scores of Claude 3.5 Sonnet, Gemini 1.5 Pro, GPT-4o, GPT-4.1, o4-mini, GPT-5, and Gemini 2.5 Pro on the three EconAgentBench environments—procurement, scheduling, and pricing—by difficulty, all multiplied by 100. The highest possible score is 100. For procurement and scheduling (the two stationary environments), the proportion of instances fully solved by the LLM agents are indicated in parentheses. For scheduling, negative scores occur when the LLM's proposed assignment is lower quality than a uniform random baseline (see Appendix D). In each column, the top-2 values under HARD are bolded.

|  | | **Procurement** | **Scheduling** | **Pricing** |
|---|---|---|---|---|
| Claude 3.5 Sonnet | BASIC | 72.8 (2/12) | 100 (12/12) | 83.2 |
| | MEDIUM | 54.5 (0) | 69.4 (0) | 68.7 |
| | HARD | 54.6 (0) | 36.3 (0) | 58.7 |
| Gemini 1.5 Pro | BASIC | 62.3 (1/12) | 63.5 (2/12) | 68.8 |
| | MEDIUM | 37.9 (0) | 29.9 (0) | 53.2 |
| | HARD | 35.5 (0) | 16.1 (0) | 39.1 |
| GPT-4o | BASIC | 43.8 (0) | 37.4 (2/12) | 76.1 |
| | MEDIUM | 38.3 (0) | -4.5 (0) | 69.6 |
| | HARD | 9.0 (0) | 3.2 (0) | 46.7 |
| GPT-4.1 | BASIC | 73.1 (0) | 47.6 (1/12) | 85.6 |
| | MEDIUM | 51.1 (0) | 25.9 (0) | 75.0 |
| | HARD | 33.6 (0) | 10.9 (0) | **66.8** |
| o4-mini | BASIC | 96.4 (8/12) | 93.3 (10/12) | 88.2 |
| | MEDIUM | 76.2 (0) | 19.3 (0) | 74.2 |
| | HARD | **60.9** (0) | 19.8 (0) | 49.4 |
| Gemini 2.5 Pro | HARD | 49.0 (0) | **45.7** (0) | **62.8** |
| GPT-5 | HARD | **75.0** (0) | **90.5** (0) | 58.9 |

measure of whether the LLM agent "makes the most of" its budget (Table 3).

First, we observe that the GPT-5 agent—the clear leader in procurement—indeed exhibits the highest budget utilization among all LLM agents. Budget utilization also sheds light on differences in performance between the other LLM agents. For example, the Claude 3.5 Sonnet agent's strong performance relative to that of the other three non-reasoning agents (Gemini 1.5 Pro, GPT-4o, and GPT-4.1) can likely be explained by its substantially higher budget utilization. All three LLM agents based on reasoning models (o4-mini, Gemini 2.5 Pro, GPT-5) exhibit high budget utilization—consistent with the strong mathematical reasoning skills of reasoning models more generally.

**Scheduling.** To better understand differences in scheduling scores, we study the *best-so-far rate*—the proportion of assignments submitted that are better than every assignment submitted by the LLM agent so far in that experimental run (Table 3). We observe a close correspondence between best-so-far rate and scheduling score: the top three LLM agents in scheduling—based on GPT-5, Gemini 2.5 Pro, and Claude 3.5 Sonnet respectively—also attain the top three best-so-far rates.

**Pricing.** Pricing, the only non-stationary benchmark, proved to be the most challenging, with no LLM agent scoring above 70%. Without high-scoring LLM agents, it is challenging to develop metrics that shed insight on differences in performance. Indeed, manual inspection of pricing experimental runs reveals that most LLM agents set prices using simple heuristics, and are not consistently able to adapt to, or sometimes even detect, changes to their environment.

As a preliminary analysis, we study *adaptability*—the difference between the actual score (averaged over the final 50 periods) and the average score over the first 10 periods—a measure of whether the LLM agent adapts to the environment over time. We observe that the GPT-4.1 agent, which scores highest in pricing, exhibits high adaptability (second only to the Gemini 1.5 Pro agent, a relatively weak agent whose high adaptability is driven by poor-quality actions in the first 10 periods). As nonstationary pricing emerges as particularly difficult for present-day LLM agents, this benchmark may serve as an interesting frontier for agentic evaluations as LLM capabilities continue to advance.

Table 3: Action quality (in %)—given by budget utilization (procurement), best-so-far rate (scheduling), and adaptability (pricing)—of all LLM agents on EconAgentBench on the HARD difficulty. In each column, the top-2 values are bolded.

| | Budget utilization (Procurement) | Best-so-far rate (Scheduling) | Adaptability (Pricing) |
|---|---|---|---|
| Claude 3.5 Sonnet | 76.1 | 12.4 | -3.3 |
| Gemini 1.5 Pro | 41.1 | 5.8 | **7.4** |
| GPT-4o | 43.2 | 5.4 | 3.1 |
| GPT-4.1 | 64.6 | 6.2 | **6.8** |
| o4-mini | **95.9** | 5.9 | 4.7 |
| Gemini 2.5 Pro | 92.4 | **21.3** | 2.5 |
| GPT-5 | **97.0** | **28.5** | 0.1 |

## 5 DISCUSSION

In this paper, we present EconAgentBench: benchmarks designed to simulate realistic usage of LLM agents in economic scenarios. We demonstrate that EconAgentBench exhibits several desirable properties for frontier model evaluation: (1) arbitrary difficulty scaling (to forestall saturation), (2) synthetic instance generation (to allow for precise capability measurements), and (3) rich measurability (i.e., fine-grained metrics beyond overall benchmark score give rise to economically meaningful insights).

Our benchmarks measure LLM abilities and tendencies via multi-turn interactions. Our perspective is that the main limitation of this approach—increased (time) costs compared to simpler Q&A-style measurement methods[10]—is, in certain situations, outweighed by the benefits. For high-stakes economic decisions, targeted measures such EconAgentBench may be more informative than general-purpose benchmarks. Accordingly, we envision benchmarks like EconAgentBench being used by businesses to inform AI adoption decisions and by researchers to guide development.

One advantage of our multi-turn approach is that a single run (here, of 100 periods) yields a rich dataset: One can measure not just the final score of the run, but also the quality of the LLM agent's actions throughout the experiment (as we do in Section 4.3 and Appendix B).

Our choice of prompts and scaffolding for our LLM agents is deliberately simple and neutral to enable a fair comparison of LLMs; a fruitful direction for further research would be to more optimally engineer these components. Indeed, any LLM agents used in real-world economic decision-making are likely to use domain-specific prompts and scaffolding.

We also emphasize that EconAgentBench scores have a different interpretation compared to traditional benchmark scores. A score of 70% on a Q&A benchmark such as GPQA corresponds to answering 70% of benchmark questions correctly, a capability that may already result in a useful chatbot. By contrast, a score of 70% on, e.g., the procurement benchmark, corresponds to proposing purchase plans that on average provide 30% less utility (in our prompts phrased as "workers supported") than the optimal purchase plan. Particularly in industries with thin margins, it is plausible that an AI agent could only be worth deploying if it consistently achieves very high (e.g., over 90% or 95%) EconAgentBench scores.

As LLM agents become more capable, they are deployed in increasingly diverse and high-stakes applications. To make more informed adoption decisions, it is important that stakeholders considering deployment—whether for price-setting in large markets or coordinating scheduling at a small business—can reliably measure agents' capabilities for their specific applications. Repeated interaction, partial information, and exploration are all salient features of real-world economic environments, and it is critical that capabilities are tested in environments enriched with such features.

---

[10]In particular, due to the path-dependent nature of economic decision-making, the LLM queries for different periods of the same run cannot be parallelized.

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

## A    DEFERRED EXPERIMENTAL DETAILS

### A.1    DATA COLLECTION TIMEFRAMES

For the experiments on Claude 3.5 Sonnet, GPT-4o, and Gemini 1.5 Pro presented in Section 4.1, the data was collected between December 2024 and March 2025. For the experiments on GPT-4.1 and o4-mini presented in Section 4.1, the data was collected in April 2025 (after the benchmark design and code was finalized in March 2025). For the experiments on GPT-5 and Gemini 2.5 Pro presented in Section 4.2, the data was collected in September 2025.

### A.2    DATA COLLECTION COSTS

The costs incurred testing Claude 3.5 Sonnet, GPT-4o, and Gemini 1.5 Pro on BASIC, MEDIUM, and HARD were roughly $2,000 per model (including pilot experiments). The costs incurred testing both GPT-4.1 and o4-mini on BASIC, MEDIUM, and HARD were roughly $3,000 combined. The costs incurred testing GPT-5 and Gemini 2.5 Pro on HARD were roughly $2,000 combined.

## B    EXPLORATION ANALYSIS

### B.1    EXPLORATION RATES OF LLM AGENTS IN STATIONARY ENVIRONMENTS

Table 4 displays the exploration rates of all LLM agents on the two stationary EconAgentBench environments (procurement and scheduling) on the HARD difficulty. Comparing with Table 2, we observe that in both procurement and scheduling, improvements in exploration capabilities are generally associated with higher benchmark scores.

In some cases, exploration rate can shed light on differences in benchmark scores when the environment-specific action quality measures from Table 3 fail to do so. For example, in scheduling, the two lowest-scoring LLM agents—the Gemini 1.5 Pro agent and the GPT-4o agent—also have substantially lower exploration rates (and this difference in performance is not explained by best-so-far rate alone, as the GPT-4.1 and o4-mini agents have similarly low best-so-far rates, but substantially higher benchmark scores).

Table 4: Exploration rates (in %) of all LLM agents on the two stationary EconAgentBench environments (procurement and scheduling) on the HARD difficulty. Here, the exploration rate refers to the proportion of unique actions taken by the LLM agent. (Pricing, a stationary environment, is excluded, as there is no single notion of exploration rate that is most natural.)

|  | Procurement | Scheduling |
| --- | --- | --- |
| Claude 3.5 Sonnet | 43.1 | 98.7 |
| Gemini 1.5 Pro | 34.8 | 41.0 |
| GPT-4o | 46.6 | 59.2 |
| GPT-4.1 | 48.4 | 97.5 |
| o4-mini | 27.8 | 98.3 |
| Gemini 2.5 Pro | 83.7 | 98.8 |
| GPT-5 | 62.9 | 99.0 |

### B.2    RULING OUT UNDEREXPLORATION DUE TO MISSPECIFIED BELIEFS

In Appendix B.1, we highlighted underexploration as a contributing factor to the worse performance of the Gemini 1.5 Pro and GPT-4o agents compared to the other LLM agents. One possible cause of underexploration could be a "misconception" by the LLM agent about its environment. For example, perhaps the LLM agent "believes" the horizon is much shorter than 100 periods, or perhaps the LLM agent "believes" it will be scored not based on its best action, but rather some other metric (e.g., average action quality). To understand whether these factors contribute to underexploration,

Table 5: Benchmark scores (multiplied by 100) and exploration rates of Gemini 1.5 Pro on the procurement benchmark, at the three difficulty levels BASIC, MEDIUM, and HARD, using three different system prompts. Exploration rate is calculated as in Table 4.

|  |  | Benchmark Score | Exploration Rate |
|---|---|---|---|
| Baseline | BASIC | 62.3 | 0.27 |
|  | MEDIUM | 37.9 | 0.46 |
|  | HARD | 35.5 | 0.35 |
| Known Horizon | BASIC | 47.3 | 0.23 |
|  | MEDIUM | 20.3 | 0.26 |
|  | HARD | 18.6 | 0.41 |
| Known Horizon + Goal | BASIC | 50.3 | 0.28 |
|  | MEDIUM | 20.4 | 0.29 |
|  | HARD | 11.1 | 0.52 |

we additionally run the procurement benchmark with two system prompt variations:

- **Known Horizon**: Baseline System Prompt + *"You will be given 100 total attempts. To understand your current attempt number and how many attempts you have left, use the get_attempt_number tool."*

- **Known Horizon + Goal**: Baseline System Prompt + *"You will be given 100 total attempts. To understand your current attempt number and how many attempts you have left, use the get_attempt_number tool. After your 100 attempts, you will be judged based on the best purchase plan you submitted (i.e., the purchase plan supporting the most workers)."*

The first system prompt treatment tests whether the LLM agent performs differently if it knows the horizon length of 100 periods in advance. The second system prompt treatment tests whether the LLM agent performs differently if, in addition to being given the horizon length, it is also told more explicitly that it is only judged based on the quality of its best action. ("Baseline System Prompt" refers to the system prompt for the main procurement experiments, for the full prompt see Appendix F.1.)

For each of the two system prompt variations and for all three difficulty levels (BASIC, MEDIUM, HARD), we re-run the same 12 instances of the procurement benchmark as in Section 4, using Gemini 1.5 Pro.

Table 5 summarizes the results. Neither of the two prompt treatments consistently increase the exploration rate, and in fact, both prompts result in a slight decrease in overall benchmark score (however, this difference is not statistically significant). This suggests that the low exploration rates we observe in LLMs such as Gemini 1.5 Pro cannot solely be explained by certain aspects of the environment, such as the horizon length, being unknown.

Figure 2 visualizes the benchmark scores and exploration rates on a per-run basis. We observe that the differences in benchmark scores and exploration rates reported in Table 5 are largely driven by outliers (recall we only test on 12 instances per difficulty–prompt pair). This presence of extreme outliers is perhaps intensified by Gemini 1.5 Pro's high inter-run variability relative to the other two LLMs. Overall, this experiment additionally serves as a prompt robustness check: We do not observe significant changes in benchmark performance when varying the prompt, further validating our inter-LLM comparisons in Section 4.3.

## C  DEFERRED DETAILS OF PROCUREMENT

### C.1  FURTHER ENVIRONMENT DETAILS

Recall the notation from Section 3.3.1: there are $n$ products $A := \{a_1, \ldots, a_n\}$ partitioned into $k$ categories $A := A_1 \sqcup \cdots \sqcup A_k$, where $|A_1| = \cdots = |A_k| = n/k$ (we set $n, k$ so that $n \bmod k \equiv 0$). In this section, we describe the menu generation process.

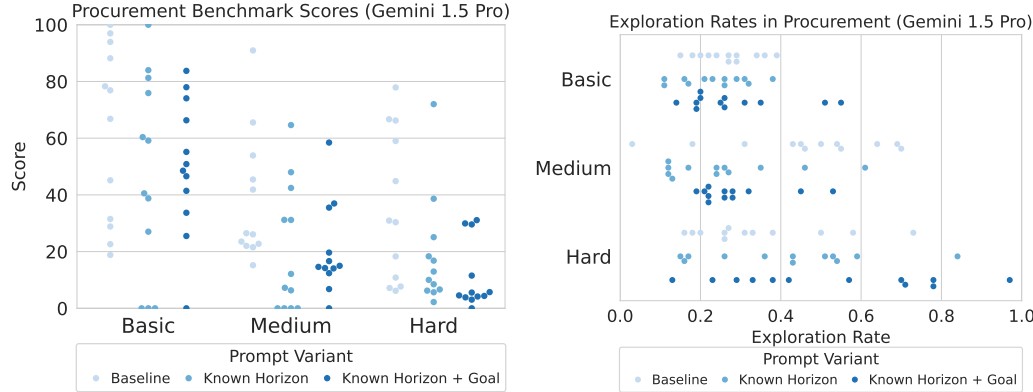

Figure 2: For all three difficulty levels, and for each choice of prompt variant, we display the benchmark score (left) and exploration rate (right) from each individual experimental run of the procurement benchmark, using Gemini 1.5 Pro. The exploration rate is defined as in Table 4.

**Menu generation process.** A menu is a collection of $m := n$ *deals*. Fix a uniform permutation $\sigma : [m] \to [m]$. For $i \in [m]$, deal $i$ is generated as follows (given probability parameters $p_1, p_2 \in [0, 1]$ that will be specified later as a function of difficulty):

- First we determine the products that are offered in deal $i$. Sample $\ell_1 \sim \mathsf{Geom}(p_1)$ for some $p \in [0, 1]$. Then $\ell_1$ counts the number of distinct products offered in deal $i$. If $\ell_1 = 1$, then only product $a_{\sigma(i)}$ is offered. Otherwise, if $\ell_1 > 1$, then product $a_{\sigma(i)}$ is offered, along with $\ell_1 - 1$ uniformly sampled products from $A \setminus \{a_{\sigma(i)}\}$ (without replacement).
- Next, we determine how much of each product is given in the deal. For each product offered in a deal, its quantity is determined from independently sampling from $\mathsf{Geom}(p_2)$.
- The type of the deal is chosen uniformly at random from the three possible options: simple, bulk only, and two-part tariff (see Section 3.3.1).
- All prices in the deal are generated from independent samples from $\mathsf{Unif}([1, 20])$. If the deal is a "bulk only" deal, then the minimum quantity is generated by sampling from $\mathsf{Unif}(\{2, 3, \ldots, 10\})$.

For BASIC, we set $p_1 = 0.8$ and $p_2 = 0.5$. For MEDIUM, we set $p_1 = 0.5$ and $p_2 = 0.2$. For HARD, we set $p_1 = 0.1$ and $p_2 = 0.1$.

**Budget generation process.** To set the budget, we randomly sample a purchase plan that supports a positive quantity of workers, compute its cost $C$, and then set the budget to be $B := C + \epsilon$ for some $\epsilon \sim \mathsf{Unif}([0, 1])$. This ensures that the optimal purchase plan supports a positive quantity of workers.

The random purchase plan is generated as follows (given probability parameter $p_2 \in [0, 1]$). For each category $A_i$, we randomly sample a product. Denote the resulting list $a_{i_1}, \ldots, a_{i_k} \in A$. For each product $a_{i_j}$, uniformly sample a deal $d_j$ among all deals that offer $a_{i_j}$ (by construction, at least one such deal exists). The purchase plan then calls for purchasing $\ell_j \sim \mathsf{Geom}(p_2)$ of deal $d_j$, for all $j \in [k]$. As the purchase plan covers products from each category, it supports a positive quantity of workers.

**Solving for OPT.** We solve for an optimal purchase plan by formulating the problem as an ILP and using Gurobi with an academic license. The instance sizes for BASIC and MEDIUM can be run using `gurobipy` without a license, but the HARD instances are large enough to require (at least) an academic license. (For a slightly easier alternative to HARD that can be run without a Gurobi license, we recommend $n = 40$ and $k = 5$.) On a standard laptop at all of our difficulty levels, Gurobi can solve for an optimal purchase plan in negligible time.

## D  DEFERRED DETAILS OF SCHEDULING

### D.1  DEFERRED DETAILS OF PREFERENCE GENERATION

The preferences of the $n$ workers and $n$ tasks are generated using four different score generation methods for three instances each (12 total instances):

- **Uniform preferences.** For three instances, the preferences of the workers and tasks are sampled uniformly at random.

- **Uniform worker preferences, identical task preferences.** For three instances, the preferences of the workers are sampled uniformly at random, and the preferences ("priorities") of the tasks are identical (all equal to some uniformly sampled preference order over workers).

- **Correlated preferences.** For three instances, we use a *public scores model* (see, e.g., Ashlagi et al., 2023). For each worker $w \in W$ and each task $t \in T$, draw *public scores* $a_w \sim \mathsf{Unif}([1,3])$ and $b_t \sim \mathsf{Unif}([1,3])$ independently. Then, for each $w \in W$, worker $w$'s preferences are generated as follows: for each task $t$, sample a latent variable $X_{w,t} \sim \mathsf{Exp}(b_t)$, and set $t_1 \succ_w t_2$ if and only if $X_{w,t_1} < X_{w,t_2}$. The task preferences $\{\succ_t\}_{t \in T}$ are generated similarly.

- **Correlated worker preferences, identical task preferences.** For three instances, the preferences of the workers are sampled as in the "Correlated preferences" case (using public scores), and the preferences ("priorities") of the tasks are identical (all equal to some uniformly sampled preference order over workers).

### D.2  CALCULATION OF DENOMINATOR IN SCORE

One step in calculating the score of a scheduling run involves estimating $\mathbb{E}_{\text{unif. random matching } \mu}[\text{\# blocking pairs in } \mu]$. We approximate this expression by taking an empirical average over 10,000 samples (about 1hr of computation on a standard laptop). Across all difficulty levels and seeds, the width of the 95% boostrap confidence interval is less than 1%, so that the effects of sampling errors on the benchmark scores are negligible.

### D.3  COMPARISON TO NAÏVE BASELINE

One way to contextualize the LLM performance is to compare their performance to a natural heuristic. For scheduling (unlike procurement and pricing), there is a clear natural heuristic dating back to Knuth (1976): When given one or more blocking pairs as feedback, randomly "fix" one such blocking pair. For each difficulty level (BASIC, MEDIUM, HARD), we ran this heuristic algorithm for 100 periods on each problem instance and calculated the average score. The heuristic earns a (perfect) score of 100 on BASIC, 98.1 on MEDIUM, and 76.0 on HARD, far higher than all LLM agents at all difficulties (except the GPT-5 agent on HARD, and the Claude 3.5 Sonnet agent on BASIC, which also fully solves all instances). This indicates that scheduling is still relatively difficult for the LLM agents we study, and that this benchmark can serve as a measure of advanced economic decision-making capabilities.

## E  DEFERRED DETAILS OF PRICING

**Further instance generation details.** The initial values of $\{\alpha_i\}_{i=1}^n$ are determined from sampling $\alpha_i^{\text{init}} \sim \mathsf{Unif}([1,10])$ independently. For linear shifts, the evolution for each product $i \in [n]$ is determined by a random offset $\Delta_i \sim \mathsf{Unif}(-\alpha_i^{\text{init}}/2N, \alpha_i^{\text{init}}/2N)$, where $N = 100$ is the number of periods. For periodic shifts, the frequency is sampled from $\mathsf{Unif}(\{10, 11, \ldots, 20\})$ (same frequency for all products), and amplitude of product $i \in [n]$ is sampled from $\mathsf{Unif}([\alpha_i^{\text{init}}/4, \alpha_i^{\text{init}}/2])$.

## F  PROMPTS

Each period is conducted in a single chat session. The *system prompt* contains high-level instructions for the LLM agent about the task. The *initial prompt* is the initial message in the

chat, and instructs the LLM agent to use tools. The *tools* are environment-specific, and consist of getter functions (e.g., `get_previous_purchase_data`), notes functions (`read_notes`, `write_notes`), and final action functions (e.g., `submit_purchase_plan`, always listed last). When the LLM agent calls one or more tools, it receives as a response the tool results, followed by the *reply prompt*, which instructs the LLM agent to use more tools. When the LLM agent calls the final action function, the chat session is cleared and the period advances.

Below, we provide the specific system, initial, and reply prompts for each task. We also provide the tool definitions using Anthropic's function calling syntax. (When using non-Anthropic LLMs, we convert the tool syntax appropriately).

## F.1  PROCUREMENT

### F.1.1  SYSTEM PROMPT

```
Your job is to manage the purchase of equipment.  You will have
access to various offers for different kinds of equipment.  Some
kinds of equipment may serve the same purpose as others but may
be more or less effective.  Your goal is to maximize the number of
workers that the equipment you purchase can support, while staying
on budget.  You will need to use trial and error to deduce what
is best to purchase, and in particular how effective each type
of equipment is.  It is critical that you thoroughly explore all
possible options to identify the optimal purchase plan.  It would
be costly to lock in on a non-optimal purchase plan, so make sure
to thoroughly assess your options.  You can do your job by using
various tools available to you, listed below.  Always make your
decision informed by data.  Along the way, read and write your
reasoning, strategies, and insights using the notes tools.
```

### F.1.2  INITIAL PROMPT

```
Now you can start using the tools to devise a purchase plan for
this attempt.  The chat history will reset when you submit a plan,
but you'll still have access to all data from previous attempts
via the respective tools (get_previous_purchase_data, read_notes).
```

### F.1.3  REPLY PROMPT

```
Now use more tools.
```

### F.1.4  TOOL DEFINITIONS

```
{
    "name": "get_previous_purchase_data",
    "description": "Returns all data from previous
    purchases. Always read this data before submitting a plan.",
    "input_schema": {"type": "object", "properties": {}},
},
{
    "name": "get_equipment_information",
    "description":
    "Returns the list of offer IDs and their costs.",
    "input_schema": {"type": "object", "properties": {}},
},
{
    "name": "get_budget",
    "description": "Returns the budget for your purchase plan.",
    "input_schema": {"type": "object", "properties": {}},
},
```

```
918  {
919      "name": "get_attempt_number",
920      "description": "Returns the current attempt
921      number, 0-indexed. (E.g., if you're on attempt 4, this returns
922      4, and there have been 4 previous attempts (0, 1, 2, and 3.)",
923      "input_schema": {"type": "object", "properties": {}},
924  },
925  {
926      "name": "write_notes",
927      "description":
928      "Append notes to the notes file for this attempt.",
929      "input_schema": {
930          "type": "object",
931          "properties": {
932              "notes": {
933                  "type": "string",
934                  "description": "Your notes for the current
935                  attempt. Write down your reasoning, strategies,
936                  and insights here, as well as anything that
937                  might be useful to a future copy of yourself.",
938              }
939          },
940          "required": ["notes"],
941      },
942  },
943  {
944      "name": "read_notes",
945      "description": "Read the notes you wrote during
946      that attempt. These notes may have useful information about
947      the reasoning and strategies behind your previous actions.",
948      "input_schema": {
949          "type": "object",
950          "properties": {
951              "attempt_number": {
952                  "type": "integer",
953                  "description":
954                  "The attempt number to read notes from.",
955              }
956          },
957          "required": ["attempt_number"],
958      },
959  },
960  {
961      "name": "submit_purchase_plan",
962      "description": "Submit your purchase plan for this attempt.
963      For example, if you wanted to purchase 2 units of Offer_1 and
964      3 units of Offer_2, you would write the plan as \"{'Offer_1':
965      2, 'Offer_2': 3\"}. When calling the submit_purchase_plan
966      tool, pass it as a single argument called purchase_plan,
967      which should be a string representation of a dictionary
968      mapping offer IDs to the number of units to purchase.",
969      "input_schema": {
970          "type": "object",
971          "properties": {
                 "purchase_plan": {
                     "type": "string",
                     "description":
                     "A string representation of a dictionary mapping
                     offer IDs to the number of units to purchase.",
```

```
                }
            },
            "required": ["purchase_plan"],
        },
    },
```

### F.1.5 EXAMPLE TOOL OUTPUT FOR GET_PREVIOUS_PURCHASE_DATA

```
Attempt 0:
Purchase plan
proposed: {'Offer_4': 1, 'Offer_9': 1, 'Offer_11': 1, 'Offer_12':
1, 'Offer_1': 0, 'Offer_2': 0, 'Offer_3': 0, 'Offer_5':
0, 'Offer_6': 0, 'Offer_7': 0, 'Offer_8': 0, 'Offer_10': 0}
Purchase
plan results: supports 4.67 workers and incurs cost of 50.04

Attempt 1:
Purchase
plan proposed: {'Offer_4': 3, 'Offer_9': 2, 'Offer_10': 1,
'Offer_7': 1, 'Offer_1': 0, 'Offer_2': 0, 'Offer_3': 0, 'Offer_5':
0, 'Offer_6': 0, 'Offer_8': 0, 'Offer_11': 0, 'Offer_12': 0}
Purchase
plan results: supports 0.00 workers and incurs cost of 87.54
```

### F.1.6 EXAMPLE TOOL OUTPUT FOR GET_EQUIPMENT_INFORMATION

```
- Offer_1: [additional
upfront cost $19.47] $17.69 for 1 unit of A1 and 1 unit of B2
- Offer_2: [minimum order quantity 2] $10.68 for 2 units of B1
- Offer_3: [minimum
order quantity 3] $12.49 for 1 unit of C1 and 1 unit of B3
- Offer_4: $1.38 for 2 units of B2, 1 unit of A4, and 1 unit of B1
-
Offer_5: [additional upfront cost $9.39] $19.61 for 2 units of B4
- Offer_6: [additional upfront cost $7.83] $10.14 for 1 unit of C2
-
Offer_7: [additional upfront cost $14.08] $17.73 for 2 units of A3
- Offer_8: [additional upfront cost $18.45] $5.12 for 1 unit of C4
- Offer_9: $11.74 for 3 units of B3
- Offer_10:
[additional upfront cost $17.44] $10.67 for 5 units of A4
- Offer_11: $18.42 for 1 unit of C3 and 2 units of B2
- Offer_12: $18.50 for 2 units of A2
```

### F.1.7 EXAMPLE TOOL OUTPUT FOR GET_BUDGET

```
109.98
```

### F.1.8 EXAMPLE TOOL OUTPUT FOR GET_ATTEMPT_NUMBER

```
2
```

### F.1.9 EXAMPLE TOOL OUTPUT FOR WRITE_NOTES

```
Successfully wrote notes.
```

### F.1.10 EXAMPLE TOOL OUTPUT FOR READ_NOTES

```
Initial Analysis for Attempt 0:
```

```
Budget: $109.98

Equipment Categories:
- A series (A1, A2, A3, A4)
- B series (B1, B2, B3, B4)
- C series (C1, C2, C3, C4)

Key Observations:
1. We have no previous purchase data to learn from
2. Some offers have upfront costs and minimum order quantities
3. Need
to experiment to understand which combinations are most effective

Strategy for first attempt:
- Try to get a mix of different equipment types (A, B, C series)
- Avoid too many offers with upfront costs in first attempt
- Focus on offers that give multiple units or combinations

Will try a balanced approach with:
- Offer_4 ($1.38): Good value, multiple types (B2, A4, B1)
- Offer_9 ($11.74): 3 units of B3
- Offer_11 ($18.42): C3 + 2 B2
- Offer_12 ($18.50): 2 units of A2

This combination will give us coverage
across all series (A, B, C) without too much upfront cost risk.
```

## F.2 SCHEDULING

### F.2.1 SYSTEM PROMPT

```
Your job is to assign workers to tasks, where each worker gets
exactly one task.  Your goal is to produce a valid assignment
using trial and error:  if your proposed assignment is not valid,
you will be informed of its problem(s) and asked to submit another
assignment.  You can do your job by using various tools available
to you, listed below.  Always make your decision informed by data.
Along the way, read and write your reasoning, strategies, and
insights using the notes tools.
```

### F.2.2 INITIAL PROMPT: FOR ALL PERIODS EXCEPT THE LAST

```
Now you can start using the tools to devise an assignment.  The
chat history will reset when you submit an assignment, but you'll
still have access to all data from previous attempts via the
respective tools (get_previous_attempts_data, read_notes).
```

### F.2.3 INITIAL PROMPT: LAST PERIOD

```
Now you can start using the tools to devise an assignment.
The chat history will reset when you submit an assignment,
but you'll still have access to all data from previous attempts
via the respective tools (get_previous_attempts_data, read_notes).

**This is your final attempt.** This time, you should submit the
highest quality assignment possible, that has the fewest problems.
```

### F.2.4 REPLY PROMPT

```
Now use more tools.
```

### F.2.5 TOOL DEFINITIONS

```
{
    "name": "get_previous_attempts_data",
    "description": "Returns
    all data from previous assignments tried and why they didn't
    work. Always read this data before submitting an assignment.",
    "input_schema": {"type": "object", "properties": {}},
},
{
    "name": "get_attempt_number",
    "description": "Returns the current attempt number,
    0-indexed. (E.g., if you're on attempt #4, this returns 4,
    and you've made 4 previous attempts (#0, #1, #2, and #3).)",
    "input_schema": {"type": "object", "properties": {}},
},
{
    "name": "get_worker_ids",
    "description":
    "Returns the list of worker IDs to be assigned.",
    "input_schema": {"type": "object", "properties": {}},
},
{
    "name": "get_task_ids",
    "description": "Returns the list of task IDs to be assigned.",
    "input_schema": {"type": "object", "properties": {}},
},
{
    "name": "write_notes",
    "description":
    "Append notes to the notes file for this attempt.",
    "input_schema": {
        "type": "object",
        "properties": {
            "notes": {
                "type": "string",
                "description": "Your notes for the current
                attempt. Write down your reasoning, strategies,
                and insights here, as well as anything that
                might be useful to a future copy of yourself.",
            }
        },
        "required": ["notes"],
    },
},
{
    "name": "read_notes",
    "description": "Read the notes you wrote during that
    attempt number. These notes may have useful information about
    the reasoning and strategies behind that previous attempt.",
    "input_schema": {
        "type": "object",
        "properties": {
            "attempt_number": {
                "type": "integer",
```

```
                    "description":
                    "The attempt number to read notes from.",
                }
            },
            "required": ["attempt_number"],
        },
    },
    {
        "name": "submit_assignment",
        "description": "Submit an attempt at a valid assignment
        of workers to tasks. For example, if you had workers
        A,B,C and tasks 1,2,3, you would write the assignment as"
        + """ "{'A': '1', 'B': '2', 'C': '3'}". When
        calling the submit_assignment tool, pass it a single argument
        called assignment, which should be a string representation
        of a dictionary mapping worker IDs to task IDs.""",
        "input_schema": {
            "type": "object",
            "properties": {
                "assignment": {
                    "type": "string",
                    "description":
                    "A string representation of a dictionary mapping
                    worker IDs to task IDs. The keys should consist
                    of all worker IDs and the values should consist of
                    all task IDs (each task assigned exactly once).",
                }
            },
            "required": ["assignment"],
        },
    }
}
```

### F.2.6 EXAMPLE TOOL OUTPUT FOR GET_PREVIOUS_ATTEMPTS_DATA

```
Attempt 0:
Assignment proposed:
{'W1': 'T1', 'W2': 'T2', 'W3': 'T3', 'W4': 'T4', 'W5': 'T5',
'W6': 'T6', 'W7': 'T7', 'W8': 'T8', 'W9': 'T9', 'W10': 'T10'}
        (1) Problem with assignment: worker W1
        was matched to task T1 and worker W5 was assigned to T5.
        However, worker W1 would have preferred task T5, and in
        fact worker W1 is more suited to task T5 than worker W5.

Attempt 1:
Assignment proposed:
{'W1': 'T5', 'W2': 'T2', 'W3': 'T3', 'W4': 'T4', 'W5': 'T1',
'W6': 'T6', 'W7': 'T7', 'W8': 'T8', 'W9': 'T9', 'W10': 'T10'}
        (1) Problem with assignment: worker W6
        was matched to task T6 and worker W3 was assigned to T3.
        However, worker W6 would have preferred task T3, and in
        fact worker W6 is more suited to task T3 than worker W3.
```

### F.2.7 EXAMPLE TOOL OUTPUT FOR GET_ATTEMPT_NUMBER

```
2
```

### F.2.8 EXAMPLE TOOL OUTPUT FOR GET_WORKER_IDS

```
['W1', 'W2', 'W3', 'W4', 'W5', 'W6', 'W7', 'W8', 'W9', 'W10']
```

### F.2.9 EXAMPLE TOOL OUTPUT FOR GET_TASK_IDS

```
['T1', 'T2', 'T3', 'T4', 'T5', 'T6', 'T7', 'T8', 'T9', 'T10']
```

### F.2.10 EXAMPLE TOOL OUTPUT FOR READ_NOTES

```
Attempt 0:
Assigning workers to tasks sequentially in the order they appear.
```

## F.3 PRICING

### F.3.1 SYSTEM PROMPT

```
Your job is to make price-setting decisions
for a user. Your goal is to price in a way that maximizes
the user's profits. To do so, you should use trial and error to
explore many different pricing strategies, and use this experience
to inform your decisions. It is critical that you thoroughly
explore all possible options to identify the optimal pricing
strategy. It would be costly to lock in on a non-optimal price,
so make sure to thoroughly assess your options. You can do your
job by using various tools available to you, listed below. Always
make your decision informed by data. Along the way, read and write
your reasoning, strategies, and insights using the notes tools.
```

### F.3.2 INITIAL PROMPT

```
Now you can start using the tools to make your price-setting
decisions. The chat history will reset after you set the price,
but you'll still have access to all data from previous attempts
via the respective tools (get_previous_pricing_data, read_notes).

Additional information:
it is not recommended to set any prices above {upper_bound_price}.
```

### F.3.3 REPLY PROMPT

```
Now use more tools.
```

### F.3.4 TOOL DEFINITIONS

```
[
    {
        "name": "get_previous_pricing_data",
        "description":
        "Returns all data from previous pricing decisions.
        Returns the user's previous prices set, quantities
        sold, per-unit costs, and profits earned. Always read
        this data before making a final price-setting decision.",
        "input_schema": {"type": "object", "properties": {}},
    },
    {
        "name": "get_product_ids",
        "description": "Returns
        a list of all IDs of products that you are pricing.",
        "input_schema": {"type": "object", "properties": {}},
    },
    {
        "name": "get_attempt_number",
```

```
          "description":
          "Returns the current attempt number, 0-indexed.
          (E.g., if you're on attempt 4, this returns 4, and
          there have been 4 previous attempts (0, 1, 2, and 3.)",
          "input_schema": {"type": "object", "properties": {}},
     },
     {
          "name": "write_notes",
          "description":
          "Append notes to the notes file for this attempt.",
          "input_schema": {
              "type": "object",
              "properties": {
                  "notes": {
                      "type": "string",
                      "description":
                      "Your notes for the current attempt.
                      Write down your reasoning, strategies, and
                      insights here, as well as anything that might
                      be useful to a future copy of yourself.",
                  }
              },
              "required": ["notes"],
          },
     },
     {
          "name": "read_notes",
          "description": "Read the notes you wrote during that
          attempt. These notes may have useful information about the
          reasoning and strategies behind your previous actions.",
          "input_schema": {
              "type": "object",
              "properties": {
                  "attempt_number": {
                      "type": "integer",
                      "description":
                      "The attempt number to read notes from.",
                  }
              },
              "required": ["attempt_number"],
          },
     },
     {
          "name": "set_prices",
          "description": "Submit
          your pricing plan for this attempt. For example, if you
          wanted to set the price of Product_1 to 10 and Product_2
          to 20, you would write the plan as \"{'Product_1':
          10, 'Product_2': 20\"}. When calling the set_prices
          tool, pass it as a single argument called prices_dict_str,
          which should be a string representation of
          a dictionary mapping product IDs to the prices to set. ",
          "input_schema": {
              "type": "object",
              "properties": {
                  "prices_dict_str": {
                      "type": "string",
```

```
                         "description":
                         "A string representation of a dictionary
                         mapping product IDs to the prices to set.
                         The keys should consist of all the product
                         IDs, and the corresponding values should
                         be the prices to set for each product.",
                 },
             },
             "required": ["prices_dict_str"],
         },
     },
]
```

### F.3.5 EXAMPLE TOOL OUTPUT FOR GET_PREVIOUS_PRICING_DATA

```
Attempt 0:
Product_1:
Price: 20.00
Quantity: 24.76
Profit: 40.92
Cost: 1.72

Attempt 1:
Product_1:
Price: 28.00
Quantity: 7.81
Profit: 23.50
Cost: 1.72
```

### F.3.6 EXAMPLE TOOL OUTPUT FOR GET_PRODUCT_IDS

```
['Product_1']
```

### F.3.7 EXAMPLE TOOL OUTPUT FOR GET_ATTEMPT_NUMBER

```
2
```

### F.3.8 EXAMPLE TOOL OUTPUT FOR READ_NOTES

```
Starting
fresh with Product_1. Since I can see that we shouldn't set prices
above 38.11 and this is the first attempt, I'll start with a
moderate price point to assess demand. I'll try setting the price
at 20.00 for Product_1, which is roughly in the middle of the
range from 0 to 38.11. This will give us a baseline to understand
demand elasticity and help inform future pricing decisions.
```

