# OpenReview forum: "EconAgentBench: Economic Benchmarks for LLM Agents in Unknown Environments"
_ICLR.cc/2026/Conference — Submitted to ICLR 2026_

### Official Review · Reviewer_TW5a · 2025-10-17

**Soundness:** 2
**Presentation:** 1
**Contribution:** 2
**Rating:** 2
**Confidence:** 3

**Summary:**

```
I used LLM to fix the grammar of the Official Review, but all opinions are my own
```
This paper introduces EconAgentBench, a benchmark designed to evaluate large language models (LLMs) such as GPT-5 and Gemini in economic decision-making scenarios. The benchmark focuses on “unknown environments,” where models are not given explicit rules upfront and must learn through exploration and trial-and-error.

While the work is technically solid—it builds a benchmark, runs experiments, and presents results—it lacks clear innovation or strong justification of practical utility. The contribution feels more like a completed academic exercise than a truly impactful advancement. I would lean toward rejecting the paper.

**Strengths:**

The idea of assessing LLMs in economic contexts is interesting and timely.

**Weaknesses:**

The claimed challenge of “unknown environments” is not compelling. Having agents learn rules through exploration is already common in RL testing; this work merely places such setups in an economic context without deeper innovation on modeling real-world uncertainty (e.g., policy shifts, competition, market shocks).

**Questions:**

1. The current scenarios (procurement, scheduling, pricing) are overly simplified. Each task isolates one factor—procurement ignores supplier delays and product quality issues; pricing ignores competition and marketing effects—making the benchmark unrealistic for genuine enterprise applications.

2. The evaluation of GPT-5, Gemini, etc., stops at surface-level score comparisons (“who performs better on which task”) without deeper analysis of why. Are performance differences due to reasoning, planning, or exploration behavior? A comparative study of model strategies would strengthen the findings.

3. The difficulty scaling (simple / medium / hard) is defined by quantity expansion (e.g., 12 products → 100 products), which is a very rudimentary approach. True economic difficulty should involve multi-objective trade-offs—balancing cost, delivery time, and quality—not just scaling task size.

4. The claimed applications are vague. For enterprises, the paper doesn’t specify actionable thresholds—what score means a model is deployable, or in what contexts (small vs. large firms). For researchers, there’s no clear roadmap of future extensions—missing domains like inventory control or supply-chain coordination, or failure modes like memorization-based performance rather than genuine exploration.

5. The writing is overly academic, filled with heavy formalism (e.g., POMDPs, nested logit demand models) while glossing over key insights and takeaways. Long sections describe tool interfaces in detail without articulating what problem they actually solve. This makes the paper feel dense, repetitive, and lacking focus on its core contribution.

---

> ### Author Response · Authors · 2025-12-02
>
> This review appears to be AI-generated in a fairly low-effort way. For example, many of the “Questions” are not actually questions -- something that a human would have noticed if they had carefully checked the AI writing before posting.
>
> Many of the reviewer’s “questions” and weaknesses center on requests for extensions in various forms. The aim of our work is to design benchmarks that isolate the aspect of economic decision-making that requires optimizing under uncertainty. Introducing additional bells and whistles, such as the extensions suggested by the reviewer, would detract from the scientific merit of our work.
>
> The reviewer also writes “The evaluation of GPT-5, Gemini, etc., stops at surface-level score comparisons (“who performs better on which task”) without deeper analysis of why.” which is simply false. In Section 4.3, our main analysis section, we already do precisely what the reviewer is asking for.

---

### Official Review · Reviewer_RCvj · 2025-10-20

**Soundness:** 2
**Presentation:** 2
**Contribution:** 1
**Rating:** 2
**Confidence:** 3

**Summary:**

In this paper, the authors present EconAgentBench. They build three small synthetic simulation environments—procurement, scheduling, and pricing—each defined by a known mathematical model and an “optimal” solution that can be computed exactly. They then let various LLMs act as agents in these environments and evaluate their performance (doing so by letting the LLMs call a few predefined tools, functions that read environment data or take an action, for 100 turns). The agents can also write and read notes to themselves between turns. In procurement, the agent buys bundles of goods within a budget to maximize a known utility function; in scheduling, it matches workers to tasks and gets back “blocking pairs” as feedback; in pricing, it sets prices over time in a demand model where price sensitivity changes each period.

**Strengths:**

- The authors are pursuing an important and, in my opinion, understudied research direction. The intersection of LLMs and economics is a fruitful area where more work is needed.
- Benchmark design is methodically implemented: three clearly defined optimization problems with computable optima, synthetic instance generation, and adjustable difficulty scaling are all implemented carefully and transparently.
- The paper provides complete reproducibility—appendices specify every parameter, generation rule, and prompt, along with cost and timeline details. For benchmarking work, this rigor is valuable.
- The use of tool-based multi-turn interaction gives a nice lightweight and extensible protocol; any LLM with function-calling capability can be plugged in, which makes the benchmark easily usable by others.

**Weaknesses:**

- Lacks substantive novelty. The work offers no new algorithmic, theoretical, or methodological contribution. It aggregates three synthetic optimization problems and frames them as economic benchmarks, without proposing new scientific questions or techniques. This in my opinion does not meet ICLR’s bar for conceptual advancement.
- The introduction presents the paper as advancing “reasoning in economic environments,” but the experiments feel more like testing in-context trial-and-error on small synthetic tasks. The framing somewhat overreaches what is actually achieved. Seems pretty similar to prior work on ICL and LLMs in economic environments (see https://arxiv.org/abs/2410.08345)
- Further along lines of the above, some claims in this paper feel somewhat poorly substantiated and not demonstrating a sound expertise in this field. Examples include:
    - "Organizations increasingly delegate parts of their economic decision-making to LLMs." - could do with more justification
    - "As the capacity for LLMs to use increasingly large sets of tools advances, one could imagine augmenting LLM agents with additional tools, e.g., a (secure) Python interpreter." - it is extremely common these days for LLMs to use code interpreter tools, and has been for a while (it's offered as a default over OpenAI API). This claim feels somewhat outdated.
- Poor engagement with prior benchmarks. The paper neglects direct comparisons to major agent benchmarks such as vals.ai Finance Agent Bench, GAIA, etc. If not going to contribute novel methodology (e.g. post-training models to be better at economic policymaking, or producing some theory etc) that's fine, but places this paper in realm of pure benchmark papers where standards these days are high.
- Questionable economic modeling. The pricing task uses an inconsistent profit formula dividing by the demand-sensitivity parameter, which breaks standard economic logic and may distort the difficulty and scoring of that environment.

**Questions:**

- Can you please check the pricing task formula? The division by alpha seems odd to me but I might be missing something.
- In general, response to the weaknesses pointed out above.

---

> ### Author Response · Authors · 2025-12-02
>
> The reviewer writes that the work lacks novelty. Our contributions, and how they stand out from prior literature, are laid out in Section 2. The scientific question our paper addresses is: “How should we measure the capabilities of LLM agents that act in unknown environments?” The contribution is designing novel economic environments amenable to LLM benchmarking, and testing LLMs on these benchmarks.
>
> The reviewer also requests comparisons to some related work, e.g. “Large Legislative Models: Towards Efficient AI Policymaking in Economic Simulations”, that is only tangentially related. We thank the reviewer for pointing us to this paper and can include it in our discussion of related work, but we point out that this is far from a core weakness of our work.
>
> The reviewer writes:
> > "As the capacity for LLMs to use increasingly large sets of tools advances, one could imagine augmenting LLM agents with additional tools, e.g., a (secure) Python interpreter." - it is extremely common these days for LLMs to use code interpreter tools, and has been for a while (it's offered as a default over OpenAI API). This claim feels somewhat outdated.
> This appears to be an honest misunderstanding. We are well aware of the functionality of modern LLM tooling. This sentence is referring to the kinds of LLM agents that we test in our experiments. We are saying that in our experiments, we do not give LLMs access to code interpreter tools, but future work might find value in testing such more complex LLM agents on our benchmarks.
>
> The reviewer asked if there was a problem with the pricing task formula. We double checked and there is not a problem. As explained in Section 3.3.3, we are using a standard nested logit demand model here.
>
> Overall, the reviewer’s weaknesses presented consist of simple misunderstandings that we would have been able to clear up in an interactive rebuttal phase, and the criticism that our work lacks novelty that we believe is unfounded (not to mention, the other reviewers praise the novelty of our approach).

---

### Official Review · Reviewer_SXQv · 2025-10-28

**Soundness:** 2
**Presentation:** 2
**Contribution:** 2
**Rating:** 2
**Confidence:** 3

**Summary:**

A suite of economic environments is proposed for evaluating LLM agents, covering canonical economic problems such as procurement, scheduling, and pricing. The environments have configurable difficulties,  and a mixture of stationary and non stationary environments. While not formalised, each of the environments can be seen as a POMDP.

**Strengths:**

- Using environments as benchmarks rather than typical QA style datasets is an important and timely direction, and the work evaluates how agents operate in an (partially) unknown environment where they must reason under uncertainty and learn through interactions.

- It's good that the benchmark is abstracted away from specific prompts or implementations, and instead operates at the environment level.

- Covers stationary and non stationary settings

- Automatically configurable difficulty levels

- Efforts to prevent data contamination through randomisation and private information

**Weaknesses:**

The key weakness is the lack of a formal framing. While POMDPs are mentioned, all environments should be explicitly cast as a POMDP (or similar) to make it much clearer. The paper would significantly benefit from a consistent formal framing specifying states, actions, observations, transitions, and rewards. This would unify the three environments, clarify what information is hidden, and make uncertainty and exploration analysis more rigorous. Currently, it is quite informal, e.g. “The agent sets prices and receives feedback”.

There are just two final results tables and no plots/breakdowns of temporal evolutions, despite all being run over 100 periods.

Smaller weaknesses:
- Figure 1 is the key description figure. This should be made more generic and clearer. If environments are formalised as POMDPs, this would be easier to see. Keep the figure generic, e.g. actions (tool usage), rewards/env scores, etc. Show periods and the repetition that occurs. Currently it is not very insightful.

- The discussion/conclusion mentions "rich measurability". Currently this analysis in the paper is nascent (just table 3).

- The impact of the notes tools should be furthered studied. This changes the state/obs space by augmenting additional info, and this would be much clearer under the POMDP formalism. Reasoning under uncertainty is mentioned, but not explored explicitly.

- For each environment, a clear description of exactly how the environments prevent data contamination, e.g. the hidden and randomly initialised private info etc, should be added.

See the questions in the section below for more in depth breakdown/clarifications around the weaknesses.

Much smaller points:
- In the related work section, the proposed benchmark seem to fit within the tool calling and/or game categorisations, rather than needing a seperate definition.
- The heavy use of footnote reference style is not particularly standard in ICLR, and should be inline where possible.

**Questions:**

- Are the environments deterministic given the hidden/private parameters?

- Every environment run is over 100 periods, what do the dynamics look like over the 100 periods? Why 100 periods? Does this give enough time to "converge"? How does the number of periods relate to the difficulty level, the level of uncertainty, etc. There is much room for improvement in this area.

- In section 3.2, "Each period is conducted in a single chat session". Should this not be either "each run", or "all periods"? As I assume it means period 0..100 are all in one chat, which isn't what is currently implied by the wording. More generally, is this not saying the state is the concatenation of all previous states, in which cases the non-stationarity is questionable (see question below). This should be stated more formally.

- The one non stationary environment features a predictable pattern, which makes it simpler than other non stationary environments. Is the time period included in the state/obs space here? (or written as a note?) If so, this env is essentially just stationary.  Again, having the envs more clearly formulated as POMDPs would help make this explicit in the paper, as well as seeing the actual temporal evolution so we can see how this changes over time.

- Do the note tools help to reduce uncertainty (e.g. my making more probable outputs, reducing variability or other)? Looking at this level of information availability overtime and the effects that have would be an insightful addition, and relate back to the POMDP, and how much is observable.

- In section 3.3 - Line 164/165, "to earn a perfect score in a non-stationary environment, it suffices for the LLM agent to identify and take an optimal action once". Should this not be "stationary"?



- Pricing: Why is the success metric just over the last 50 periods, not the full 100? Additionally, when comparing the first 10 periods to the final 50 periods, does changing the sensitivity parameter alpha not change the resulting profits? How can we compare periods under different alphas?

---

> ### Author Response · Authors · 2025-12-02
>
> The reviewer writes: “The key weakness is the lack of a formal framing. While POMDPs are mentioned, all environments should be explicitly cast as a POMDP (or similar) to make it much clearer.” The claim that this is a weakness of our paper, let alone a *key* weakness, is puzzling. The contribution of our paper is to design economics-inspired benchmark environments in which to experimentally test the capabilities of LLMs. We describe the details of all of our benchmark environments in rigorous and complete detail. The reviewer’s suggestion would actively detract from the clarity of exposition.
>
> The reviewer also writes “There are just two final results tables and no plots/breakdowns of temporal evolutions”. This is also completely incorrect. Section 4.3, the main analysis section of the paper, contains breakdowns of temporal evolutions.
>
> We thank the reviewer for pointing out some minor typos and local clarity issues that we will address, but their core rejection reasons do not appear to have much substance.

---

### Official Review · Reviewer_EU7X · 2025-10-31

**Soundness:** 2
**Presentation:** 2
**Contribution:** 1
**Rating:** 2
**Confidence:** 4

**Summary:**

- The authors propose a new benchmark for evaluating the performance of LLMs in 3 economics-related tasks: procurement, scheduling and pricing. Each task allows for scalable difficulty, offers the LLM specific tools and has associated metrics that can be used to gauge model performance.
- Experimentally, the authors evaluate various LLMs including GPT-5 and Gemini 2.5 Pro on their benchmark at 3 specific difficulty levels (Basic, Medium and Hard). In particular, the paper attempts to analyze the performance of these models by examining the adaptability and exploration levels of these models.

**Strengths:**

- Examining the performance of LLM agents at economics-related tasks is likely to be a worthwhile endeavour given the growing prevalence and agency of LLM agents.
- The authors test a wide variety of models, at a non-negligible cost, which are representative of frontier model capabilities.
- The pricing task is well-motivated and a good model of a problem that someone will likely attempt to use LLMs on at some point

**Weaknesses:**

- For the procurement and scheduling tasks, the setup feels quite strange. In both cases, the uncertainty over effectiveness/preferences seems unrealistic given the presence of an oracle function that can evaluate proposed bundles/matching.
	- For procurement, could such an oracle function exists without actually testing out the purchased bundles? If testing is required, the problem should be modelled in an online manner and some measure akin to regret should be evaluated.
	- For matching, it's unrealistic to be able to know the number of blocking pairs in a matching without knowing the preferences.
		- Maybe it would make sense to see, given the preferences, how well an LLM can discover a stable matching?
	- The problems feel almost like the bandit setting with an underlying structure to rewards?
- The main benefit of using LLMs for economic decision-making over optimization algorithms seems to be in cases where there is a lot of unstructured text data. In such cases, it is unlikely that the LLM will have access to well-structured tools.
- For all tasks, it would be helpful to have better baselines to get an idea of how hard these problems are (something similar to the naive baseline mentioned in the appendix). Ideally, there would be both the performance of a simple heuristic algorithm and some notion of human performance.
- The authors mention there is significant variation between runs but there are no error bars or indication of the variability of performance. I understand the runs are expensive (especially on frontier models) but it would be helpful to have some analysis of this effect.
- The paper could do with a deeper comparison to existing work. While there may not be many economics-related benchmarks many of the tasks do seem more like mathematical optimization problems.
- It feels like these tasks are evaluating more the ability of these models to infer key parameters of a model and then reason accordingly. However, in the prompts, it doesn't feel like the models are given enough information to be able to perform this inference.

**Questions:**

- For the pricing task, is the optimal profit computed given perfect knowledge of the varying demand? If so, can any online algorithm to reach that performance with only partial knowledge?
- Have you analyzed how well the models perform at using the tools correctly/do they ever fail to generate well-formatted actions?
- Are the notes used by the models?

---

> ### Author Response · Authors · 2025-12-02
> **Response**
>
> Re Weakness 1: For procurement and scheduling, the reviewer writes “the uncertainty over effectiveness/preferences seems unrealistic”. We actually view the environment uncertainty as making the simulations more realistic, rather than less. For scheduling, for example, one can imagine a manager of a small business repeatedly asking an LLM agent to propose a shift schedule, and then asking workers if they have any swaps they’d like to make, until the matching is stable (it may be too mentally taxing to elicit complete preference lists from the workers). For procurement, the uncertainties in the effectiveness scores can reflect, e.g., uncertainties in qualities of different products (one may have to test out a product first to understand if it’s a good fit).
>
> The suggestion by the reviewer (“Maybe it would make sense to see, given the preferences, how well an LLM can discover a stable matching?”) would make for a much less interesting benchmark. The main insight of our paper is that one can uncover valuable insights by testing the capabilities of LLM agents in economic environments with prominent *unknown* components. (Especially because there is a comparative glut of e.g. mathematics benchmarks, in which the LLM is given all information upfront, like in the modification the reviewer suggests.)
>
> Re Weakness 2: The reviewer writes “In such cases, it is unlikely that the LLM will have access to well-structured tools.” Our benchmarks are based on tool use as a necessary abstraction and standardization. Considering the optimization part of the benchmarks proves to be more difficult for the LLMs than tool use syntax, we consider this concern to be second order.
>
> Re Weakness 3: The reviewer requests baselines, which we already discuss and provide in Appendix D.3.
>
> Re Weakness 6: The reviewer raises a concern that we already precisely address with an additional experiment in Appendix B.2.
>
> Finally, the reviewer requested additional error analysis (Weakness 4 -- we already do some of this, but can further build it out) and related optimization work (Weakness 5 -- again, we already do this fairly extensively in the related work, but can further build it out).
>
> Regarding the reviewer’s questions: yes, the notes tools are used by the models, and yes, the models in almost all cases use the tools correctly (and in the rare cases they don’t, the query is retried).

---

### Meta-Review · Area_Chair_c7iy · 2026-01-07

**Summary:**

The consensus to reject the paper is clear.

**Reviewer Concerns:**

There are open questions on novelty and evaluation by all reviewers.

**Reviewer Scores:**

Scores remain below the rejection bar.

---

### Decision · Program_Chairs · 2026-01-26

Reject